# Molecular Epidemiology and Evolution of Coxsackievirus A9

**DOI:** 10.3390/v14040822

**Published:** 2022-04-15

**Authors:** Hehe Zhao, Jianxing Wang, Jianhua Chen, Ruifang Huang, Yong Zhang, Jinbo Xiao, Yang Song, Tianjiao Ji, Qian Yang, Shuangli Zhu, Dongyan Wang, Huanhuan Lu, Zhenzhi Han, Guoyan Zhang, Jichen Li, Dongmei Yan

**Affiliations:** 1National Polio Laboratory, WHO WPRO Regional Polio Reference Laboratory, National Health Commission Key Laboratory for Biosecurity, National Health Commission Key Laboratory for Medical Virology, National Institute for Viral Disease Control and Prevention, Chinese Center for Disease Control and Prevention, Beijing 102206, China; catherine0185@126.com (H.Z.); zhangyong@ivdv.chinacdc.cn (Y.Z.); mr_mint1114@sina.com (J.X.); candyalbarn57@126.com (Y.S.); jtj112@163.com (T.J.); yangqian@ivdc.chinacdc.cn (Q.Y.); zhusl@ivdc.chinacdc.cn (S.Z.); yanzi1973_55@163.com (D.W.); luhuanhuan0908@163.com (H.L.); hansir8@sina.com (Z.H.); zhangguoyan0731@126.com (G.Z.); jichenli666@163.com (J.L.); 2Department for Viral Disease Control and Prevention, Shandong Center for Disease Control and Prevention, Jinan 250014, China; jianxing_wang@163.com; 3Department for Viral Disease Control and Prevention, Gansu Center for Disease Control and Prevention, Lanzhou 730000, China; ninachen0931@163.com; 4Department for Communicable Disease Control and Prevention, Xinjiang Uygur Autonomous Region Center for Disease Control and Prevention, Urumqi 830011, China; fang72718@sina.com; 5Center for Biosafety Mega-Science, Chinese Academy of Sciences, Beijing 102206, China; 6Department of Medical Microbiology, Weifang Medical University, Weifang 261053, China

**Keywords:** coxsackievirus A9, genotyping, recombination, evolutionary reconstruction

## Abstract

Nineteen CVA9 isolates were obtained between 2010 and 2019 from six provinces of mainland China, using the HFMD surveillance network established in China. Nucleotide sequencing revealed that the full-length VP1 of 19 CVA9 isolates was 906 bases encoding 302 amino acids. The combination of the thresholds of the phylogenetic tree and nucleotide divergence of different genotypes within the same serotype led to a value of 15–25%, and enabled CVA9 worldwide to be categorized into ten genotypes: A–J. The phylogenetic tree showed that the prototype strain was included in genotype A, and that the B, C, D, E, H, and J genotypes disappeared during virus evolution, whereas the F, I, and G genotypes showed co-circulation. Lineage G was the dominant genotype of CVA9 and included most of the strains from nine countries in Asia, North America, Oceania, and Europe. Most Chinese strains belonged to the G genotype, suggesting that the molecular epidemiology of China is consistent with that observed worldwide. The 165 partial VP1 strains (723 nt) showed a mean substitution rate of 3.27 × 10^−3^ substitution/site/year (95% HPD range 2.93–3.6 × 10^−3^), dating the tMRCA of CVA9 back to approximately 1922 (1911–1932). The spatiotemporal dynamics of CVA9 showed the spread of CVA9 obviously increased in recent years. Most CVA9 isolates originated in USA, but the epidemic areas of CVA9 are now concentrated in the Asia–Pacific region, European countries, and North America. Recombination analysis within the enterovirus B specie (59 serotypes) revealed eight recombination patterns in China at present, CVB4, CVB5, E30, CVB2, E11, HEV106, HEV85, and HEV75. E14, and E6 may act as recombinant donors in multiple regions. Comparison of temperature sensitivity revealed that temperature-insensitive strains have more amino acid substitutions in the RGD motif of the VP1 region, and the sites T283S, V284M, and R288K in the VP1 region may be related to the temperature tolerance of CVA9.

## 1. Introduction

The genus *Enteroviruses* of family *Picornaviridae* and order *Picornavirals* includes the species enterovirus (EV) A–L and rhinovirus (RV) A–C. Enteroviruses infecting humans are assigned to four species, EV-A–D [1]. Human enteroviruses include over 100 serotypes, comprising poliovirus, coxsackievirus, echovirus, and some newly discovered enteroviruses [2]. Coxsackievirus A9 (CVA9) is an important human pathogen belonging to EV-B of the *Picornaviridae* family and associated with a wide spectrum of clinical symptoms, including aseptic meningitis (AM), hand, foot, and mouth disease (HFMD), acute flaccid paralysis (AFP), and persistent diarrhea [3]. The viral RNA of CVA9 is approximately 7.5 kb, which contains a long open reading frame (ORF) flanked by 5’ and 3’ untranslated regions (UTRs), and the encoded polyprotein is cleaved to produce the structural and nonstructural proteins. The ORF encodes a single polyprotein of 2201 amino acids which is first cleaved into four structural proteins (VP1, VP2, VP3, and VP4) and seven non-structural proteins (2A, 2B, 2C, 3A, 3B, 3C and 3D). The structural proteins P1, 2A, 2B, and 2C are derived from P2; and 3A, 3B, 3C and 3D from P3 [4]. 

Serotype identification and genotype classification are important components of the study of the molecular evolution of enteroviruses. Genotyping analysis of CVA9 can provide a reliable theoretical basis for the study of the genetic characteristics of CVA9 and the analysis of virus evolution. Although CVA9 has been associated with a wide spectrum of clinical symptoms, there is no well-accepted classification system available for CVA9. Juhana et al. identified 12 CVA9 genotypes named I-XII by analyzing 150nt of the VP1/2A linkage region, which was the first genotyping analysis of CVA9 [5]. In 2005, Cui et al. compared the strains isolated from an outbreak of aseptic meningitis in Gansu Province of China with these 12 genotypes and found that the genetic distance was 0.21–0.40, which did not belong to any of the 12 types, and therefore, defined the strains they studied as type XIII [6]. In 1999, Oberste et al. found that the VP1 coding region carries major neutralization epitopes among the capsid proteins and is likely to be the best region for virus identification and molecular typing [7]. Sequence analysis of VP1 has been applied to the genotype classification analysis of most enteroviruses, such as EV-A71 [8], CVA16 [9], CVA6 [10], and CVB1 [11]. The proposal made by Oberste et al. to classify enterovirus genotypes and sub-genotypes according to 15–25% and 8–15% nucleotide differences, respectively, in the complete VP1 region, followed by phylogenetic analysis, has become the prevailing standard [12]. The genotyping of CVA9 based on partial coding region of VP1 has been mentioned in studies in China. For example, Zhang et al. [13] classified the CVA9 strains into 2 different branches based on partial VP1, and Xu [14] and Ma et al. [15] classified CVA9 into 3 and 4 genotypes, respectively. Although all domestic typing studies of CVA9 used the VP1 coding region, there was not an adopted and applied typing method of CVA9 for the limitations of the small-time span and the narrow geographical range of sequences.

RNA viruses have a high mutation rate due to the lack of proofreading activity during genome replication, although their genomes are relatively short [5]. It is estimated that approximately one mutation is generated per newly synthesized genome [16]. In addition to mutations, recombination is involved in EVs’ evolution. Recombination in EVs was first observed between serotypes of poliovirus in vaccine recipients, and since in a wide range of human enteroviruses [17]. Since recombination requires two different viruses to infect a cell at the same time, the recombinant strains need to be biastophilic, and phylogenetic analysis of complete enterovirus genomes has provided evidence suggesting that intraspecies recombination is a relatively frequent event in the evolution of EVs [7]. In addition, recombination may reassemble genes with different dominant characteristics into a new genome, allowing viral resistance, immune evasion, virulence evolution, etc. [18]. The genetic exchange of EVs may lead to the emergence of new strains with unknown biological characteristics, which may pose a significant risk to public health.

Statistical phylogenetic analysis is an evolutionary reconstruction analysis method combining information from different sources of the study subjects with relevant sequence data, and this analysis method is widely used in simulating various infectious disease transmission and epidemiological history reconstruction studies [19,20]. CVA9 had an endemic pattern of circulation in America between 1970 and 2005 [21] and was involved in outbreaks of aseptic meningitis in China [6,15,22] and Canada [23]. Additionally, these investigations were triggered by the huge number of hospitalized children and the attention of public health officials, not by surveillance data, because aseptic meningitis has not been classified as a notifiable disease worldwide [24]. During the transmission of CVA9, both patients and asymptomatic infected persons can act as sources of infection and spread the virus through various routes, such as the fecal–oral route, daily contact, or airborne droplets, providing theoretical feasibility for the study of spatio-temporal dynamic transmission of CVA9 [25]. Therefore, it will be important to carry out the statistical phylogenetic analysis of CVA9 to infer the temporal and spatial evolution of CVA9 with the Bayesian stochastic search variable selection (BSSVS) model in BEAST software [26]. Integration of sequences with geographical information systems allows the temporal and spatial distribution of CVA9 to be mapped across global.

Although CVA9 has been associated with causing a wide range of human diseases worldwide, there is no well-accepted classification system available for CVA9. Additionally, so far, no detailed molecular epidemiology studies are available which reveal the circulation pattern, genotype distribution, or genetic diversity of CVA9 strains, nor are there any inferring its geographic spread based on phylogenetic data. In this study, we integrated 19 isolates collected from mainland China with all complete VP1 and full-length CVA9 sequences in GenBank, resulting in a molecular evolutionary analysis of CVA9 worldwide.

## 2. Materials and Methods

### 2.1. Virus Isolation and Nucleotide Sequencing

We processed 19 stool samples of HFMD based on the HFMD surveillance network established in China according to standard procedures [27]; then the samples were inoculated into human rhabdomyosarcoma (RD) cells. RD cells were supplied by American Center for Disease Control and Prevention. Nineteen isolates were collected from Shandong (*n* = 8), Gansu (*n* = 3), Xinjiang (*n* = 4), Yunnan (*n* = 1), Hebei (*n* = 1), Shaanxi (*n* = 1), and Guangxi (*n* = 1) in mainland China between 2010 and 2019 based on the HFMD Surveillance Network established in China. Viral RNAs were extracted using a QIAamp Viral RNA Mini Kit (Qiagen, Valencia, CA, USA) following the manufacturer’s protocol. We performed reverse transcription-polymerase chain reaction (RT-PCR) to amplify the entire VP1 capsid region (906nt) using a PrimeScript One-Step RT-PCR Kit Ver.2 (TaKaRa, Dalian, China) with previously designed primers [15]. The primers used for PCR amplification and sequencing of the remaining genome in this study were designed based on the primer walking method (Appendix A). The PCR products were purified using a QIAquick PCR Purification Kit (Qiagen, Hilden, Germany), and then amplicons were bidirectionally sequenced using an ABI 3130 Genetic Analyzer (Applied Biosystems, Foster City, CA, USA).

### 2.2. Datasets Construction of Worldwide and Chinese CVA9

In addition to 19 samples in this study, sequences for genotype classification, discrete geography analyses, and recombination analysis were recruited from GenBank (data to 3 November 2021). Questionable and low-quality sequences were eliminated by TempEst analysis [28], based on (1) an error in phylogenetic inference; (2) sequences that were mislabeled and had been ascribed incorrect dates of sampling during data collation and processing; (3) the use of incorrect sampling dates for archived, reference, or vaccine virus strains. A total of 110 complete VP1 sequences were finally recruited from GenBank. After combining 19 sequences from this study and 110 sequences from GenBank together, we selected 128 complete VP1 sequences out of a total of 129 worldwide sequences for the phylogenetic analysis. For the phylogeographic analysis, we downloaded all CVA9 sequences from GenBank whose lengths were between 600 and 7500 nt; a total of 222 VP1 sequences were recruited from GenBank. Of the 19 sequences from this study and 222 partial VP1 sequences from GenBank together, we selected 165 out of a total 241 worldwide partial VP1sequences for phylogeographic analysis. For the recombination analysis of Chinese strains, 9 full-length sequences from GenBank and 19 sequences from this study were used, so a total of 28 complete sequences composed the dataset for recombination analysis.

### 2.3. Phylogenetic Analysis

Sequence data were stored as standard chromatogram format files and assembled by using Sequencher program (version 5.4.5) (GeneCodes, Ann Arbor, Michigan, USA). Sequence alignment was conducted using the ClustalW tool in MEGA (version 7.0) (Sudhir Kumar, Arizona State University, Tempe, Arizona, USA). The best nucleotide substitution models were selected using Modeltest (version 3.7). Additionally, the maximum likelihood (ML) method was used in the phylogenetic tree construction with 1000 bootstrap replicates in MEGA; then the ML tree based on the best-fit evolutionary model was exported to NWK format and imported into TempEst (version 1.5.3) [28] to confirm that the sequences under investigation contained sufficient “temporal signal” for reliable estimation. Finally, the ML tree based on the filtered complete VP1 sequences selected by TempEst with best evolutionary model was constructed to describe the CVA9 genetic diversity on a global scale.

### 2.4. Phylodynamic Analysis

To understand the spatial dynamics of CVA9 viruses, phylogeographic analyses based on 165 partial VP1 sequences (723 nt) were performed in BEAST (version 1.10) [26] to reconstruct the ancestral geographical countries, diffusion rates, and migration patterns simultaneously. The sequences for the analyses were standardized for each location and time period to minimize the effect of geographical sampling bias. Fourteen geographical locations (that is, China, USA, Australia, Poland, India, France, Tunisia, Japan, Canada, Cuba, UK, Thailand, Romania, and Russia) were selected and coded as discrete states. To estimate the diffusion rates among locations, the asymmetric substitution model with the BSSVS option was used in BEAST to infer asymmetric diffusion rates between any pairwise location state and also allow BF calculations to test for significant diffusion rates. We used a strict clock model with Bayesian skyline to estimate the rate of molecular evolution of CVA9. Output from BEAST was analyzed within the TRACER program (version 1.7.1) [29] to ensure convergence through graphical checks and adequate quality control parameters of the posterior distribution (ESS > 200). A maximum clade credibility (MCC) tree was constructed using TreeAnnotator, and the burn-in option was used to remove the first ten-percent of sampled trees, and the resulting tree was visualized by FigTree (version 1.4.4). The resulting log files were used to calculate the BF for the diffusion between discrete locations, and to extract the actual non-zero rates and mean indicators for all statistically supported routes. SpreaD3 package (version 0.9.7) was used for analysis and visualization of pathogen phylodynamic reconstructions. Significant migration pathways were summarized based on the combination of both BF > 3 and the mean indicator of >0.5. We defined the degree of rate support as follows: BF ≥ 1000 indicates decisive, 100 ≤ BF < 1000 indicates very strong support, 10 ≤ BF < 100 indicates strong support, and 3 ≤ BF < 10 indicates support.

### 2.5. Recombination Analysis

For the recombination analysis of Chinese strains, we selected the sequences of EV-B prototype strains in the GenBank. Nine full-length sequences from GenBank and nineteen sequences from this study were used to construct the neighbor-joining (NJ) phylogenetic trees of VP1, P1, P2, and P3 regions. The sequences were processed by MEGA with the best evolutionary model (GTR+G+I), and the NJ tree was constructed in MEGA (version 7) with 1000 bootstrap replicates. We initially determined the possible recombination patterns by observing the different positions of each CVA9 strain on each evolutionary tree, with the formation of different clusters, and selected one representative strain for each recombination pattern for subsequent recombination analysis. Full genomes were analyzed separately to detect recombination events using the Recombination Detection Program 4 (RDP4, version 4.46). BLAST was used to analyses the P2 and P3 regions of the Chinese strains, including the strains in this study, and full-length sequences with over 85% similarity in GenBank were obtained as potential parents. A total of seven methods implemented in RDP4 were applied, including RDP, GENECONV, 3Seq, Chimaera, SiScan, MaxChi, and LARD. Recombination detected by at least three of the seven methods with a P value cutoff 0.05 was considered as true recombination [30].

### 2.6. Assay for Temperature Sensitivity

Temperature sensitivities of CVA9 were assayed on monolayers of RD cells in 24-well plates, and temperature-sensitive and temperature–insensitive strains (HTPS-QDH11F/XJ/CHN/2011 and HTYT-ARL-AFP02F/XJ/CHN/2011, respectively) were selected as control strains [31,32]. Then, 50 μL of undiluted strains were inoculated into 24-well plates and cultured in incubators at optimal and supraoptimal temperatures for viral culture (36 °C and 39.6 °C, respectively). After absorption at 36 or at 39.5 ℃ for 1 h, the unabsorbed viruses were removed, 100 μL of maintenance medium was added to each well, and the plates were continually incubated at set temperatures separately. The 24-well plates were incubated at 36 or 39.5 °C, and the virus was harvested at 4, 8, 16, 24, 48, and 72 h post-infection. Virus isolates showing more than 2-logarithm reductions in titers at different temperatures were considered to be temperature sensitive.

### 2.7. Nucleotide Sequence Accession Numbers

The nucleotide sequence numbers for the sequences submitted to GenBank are OM885385–OM885403.

## 3. Results

### 3.1. Virus Isolation

The 19 samples were from Shandong (8), Shaanxi (1), Gansu (3), Xinjiang (4), Yunnan (1), Hebei (1), and Guangxi (1), of which three were from severe HFMD cases in Gansu and the rest were from mild cases. Nineteen CVA9 strains in this study showed a cytopathic effect (CPE) on RD cells. Sequencing of the full-length sequences showed that these isolates had 7452 bases and 2202 amino acids, with nucleotide and amino acid similarities of 80.5–97.6% and 92.3–99.6%, respectively, with the CVA9 prototype (Griggs, GenBank number: D00627.1).

### 3.2. Dataset Description

A total of 128 (include 19 sequences in this study) representative complete VP1 genome sequences worldwide were analyzed to establish the genotyping method of CVA9. The strains were isolated between the 1950s and 2019 from 12 countries, including China (*n* = 74), the UK (*n* = 5), the USA (*n* = 8), Australia (*n* = 6), France (*n* = 3), Canada (*n* = 2), Thailand (*n* = 3), Russia (*n* = 10), Cuba (*n* = 12), India (*n* = 3), Poland (*n* = 1), and Romania (*n* = 1), representing wide temporal and regional distribution (Appendix A).

### 3.3. Phylogenetic Analysis

Based on the evolutionary model of GTR+G+I and 128 complete VP1 sequences (906 nt), the maximum likelihood method was used to infer the phylogenetic evolutionary tree of CVA9 (Figure 1). Combining the phylogenetic tree with the principle that a full-length VP1 sequence nucleotide difference <15–25% can be defined as a same genotype, the CVA9 could be categorized into ten genotypes: A–J. We verified genotype classification by analyzing inter-group and intra-group distances and found that the average distance between the ten groups was 10–20%, which met the criteria of 15–25% for defining genotype by nucleotide differences in enteroviruses [33]. We named the prototype strain (Griggs) of CVA9 isolated in 1950 genotype A, which is consistent with earlier studies [15,34], and the rest were divided according to the time of isolation. Genotypes B and C were isolated in the United Kingdom in 1962 and 1979, respectively. A Romanian strain isolated in 1980; Cuban strains isolated between 1990–1993, and 2000; and a French strain isolated in 2013 belong to genotype D. Genotype E refers to a UK strain discovered in 1985. A UK strain from 1987, the Russian strains from 2012–2013, and some Indian strains all belong to genotype F. Most Chinese strains, certain French strains, Canadian strains, Australian strains, Thai strains, USA strains, and Polish strains belong to the genotype G, whereas the CVA9 strain from Australia in 1999 belongs to the genotype H. Russian strains isolated in 2012–2013, a French strain, and a Chinese strain detected in 2018 belong to genotype I. An American strain isolated in 2015 was classified as J genotype. The phylogenetic tree showed that the B, C, D, E, H, and J genotypes disappeared during virus evolution, whereas the F, I, and G genotypes showed co-circulation. Lineage G was the dominant genotype of CVA9 and included most of the strains from nine countries in Asia, North America, Oceania, and Europe. Most Chinese strains belonged to the G genotype, suggesting that the molecular epidemiology of China was consistent with that observed worldwide. However, one CVA9 strain isolated from a child with HFMD in Xinjiang of China in 2018 was closely related to Russian strains and was categorized as I genotype. It is speculated that the strain was imported from Russia. Although CVA9 has obvious geographic clustering, the multi-country and multi-period strains contained in the G genotype also suggest that there may be long-distance transmission and prevalence of CVA9 worldwide.

### 3.4. Phylodynamic Analysis

The temporal genetic analyses of CVA9 isolate sequences collected between 1962 and 2019 done with BEAST that visualize estimates of CVA9 divergence times are shown in Figure 2 (Appendix A). The maximum clade credibility (MCC) tree constructed from 165 partial VP1 region sequences (723 nt) of CVA9 has a similar topology to the phylogenetic tree, and the typing results are generally consistent. Temporal phylogenies give clues to the evolutionary rate, and the time to most recent common ancestor (tMRCA) for the whole data set of partial VP1 sequences. The partial VP1 region showed a mean substitution rate of 3.27 × 10^−3^ substitution/site/year (95% HPD range 2.93–3.60 × 10^−3^). Additionally, partial VP1 dated the tMRCA of CVA9 back to approximately 1922 (1911–1932). The Chinese CVA9 strains originated around 1978 (95% HPD: 1964–1989). The Bayesian skyline showed that the genetic diversity of nucleotides in the VP1 region of the CVA9 fraction did not change much before 1992, rose slightly between 1992 and 2003, without significant fluctuations occurring, and then began to decline rapidly, reaching a minimum in nucleotide genetic polymorphism in 2005, and began to decline after a phase increase between 2005 and 2010. Currently, it maintains a relatively stable state. The data may have created some bias in the results due to the size of the sample collection and temporal and spatial differences.

To understand the global circulation of CVA9, we reconstructed the spatial transmission patterns from 1962 to 2019, inferred from estimates of genetic diversity for 14 geographic countries (that is, the USA, Canada, Cuba, Romania, the UK, France, Tunisia, Poland, Russia, Japan, China, India, Thailand, and Australia). Based on the analysis of available CVA9 sequences, most CVA9 strains, including those in China, probably originated in the USA, whereas the current CVA9 epidemic is mostly concentrated in the Asia–Pacific and European regions. Our phylogeographic analysis indicates five significant migration pathways: (1) USA to UK; (2) USA to Tunisia; (3) USA to China; (4) USA to Thailand; (5) France to Romania (decisive support with BF ≥ 1000) (orange arrows in Figure 3). Very strongly and strongly supported diffusion rates were also indicated between numerous regions worldwide. In total, resulting diffusion rates indicate that the USA, Japan, and India played a key role in seeding the CVA9 epidemics; and Cuba, India, Poland, Thailand, Tunisia, France, and China established strong epidemiological links with multiple regions in this transmission network (Appendix A).

### 3.5. Analysis of Recombination Patterns of CVA9

Nine full-length sequences from GenBank and nineteen sequences from this study were used to do the recombination analysis of Chinese strains. Based on the sequences of all EV-B protype strains in the GenBank database and those of the CVA9 strains in China, we constructed phylogenetic trees based on VP1, P1, P2, and P3 regions, respectively (Figure 4). Both VP1 and P1 phylogenetic trees showed that the CVA9 strains in China were clustered with the CVA9 protype, which confirmed previous molecular typing results. However, the phylogenetic trees constructed based on the P2 and P3 regions showed different results. In the P2 region, China_XJ_2018_99 was clustered with the HEV86 protype strain, and some were clustered with the HEV106 protype strain. In the P3 region, they were clustered with E13, HEV106, and HEV85, suggesting that the CVA9 strains may have recombination in the P2 and P3 coding regions.

Sequences analysis of the CVA9 strains in China by RDP4 showed that there are eight recombination patterns at present (Table 1): (I) The breakpoint positions are mainly located at 3580–3780 and 4960–5890, covering parts of P2 and P3 regions of the genome; major and minor parents are CVB4 (GenBank number:KP289433) and CVB5 (GenBank number: MW015048), respectively. (II) China_HeB_2019_98 potentially had recombination events with E14 (GenBank number:KP289441) and HEV85 (GenBank number: JX898098) at position 2C. (III) The breakpoint positions strains are located at 3568–4188, covering part or all of 2A and 2B; major and minor parents are Echo7 (GenBank number:MH043132) and Echo7 (GenBank number: MH043132). (IV) The breakpoint positions are located at 1–811 and 4821–5890, covering part or all of the 5’, P2, and P3 regions. The major and minor parents in 5’ are HEV85 (GenBank number: JX898907) and Echo6 (GenBank number: KX641241), respectively. Additionally, the major and minor parents located in P2 and P3 are HEV106 (GenBank number: KX171337) and Echo11 (GenBank number: MN597951), respectively. (V) The breakpoint positions are located at 1–514 and 5495–7470, covering part or all of the 5’, P2 and P3 regions. The major and minor parents in 5’, P2, and P3 are all Echo11 (GenBank number:ky981558) and CVB3 (GenBank number: AY896763), respectively. (VI) The breakpoint positions are mainly located at 4277–7464, covering all P3 region of the genome; major and minor parents are CVB4 (GenBank number:KP289433) and CVB5 (GenBank number:MW015048), respectively. (VII) The China_XJ_2018_99 classified as G genotype was recombined with HEV85 (GenBank number: JX898907) and HEV75 (GenBank number:MW183137). (VIII) The breakpoints of the CVA9 isolates of China in the GenBank database are mainly located in P2 and P3; major and minor parents are Echo30 (GenBank number: JX976773) and HEV75 (GenBank number: AY556070) (Figure 5).

### 3.6. Temperature Sensitivity Properties of the CVA9

We selected four representative strains of CVA9 for temperature sensitivity determination based on the time and location of the isolation of the 19 strains in this study combined with genotype, three of which were from the G genotype and one from the I genotype. Four representative CVA9 strains from different clusters were compared to control strains with regard to replication capacity at a supraoptimal temperature (39.5 °C) and showed different temperature sensitivity properties. Although the replication ability of CVA9 strains varied slightly at different temperatures, differences in China_XJ_2018_99 (I) in the titer at 36 and 39.5 °C did not exceed two logarithms, whereas differences in China_YN_2014_131, China_SD_2019_323 and China_GS_2015_61 (G) did (Figure 6).

The temperature sensitivity test is a good indicator of virulence and contagiousness [35]. As previous studies have shown that the arginine–glycine–aspartic (RGD) motif found in the VP1 capsid protein of CVA9 has a role in cell entry. Analysis of the 19 CVA9 isolates found that all of them possessed the RGD motif (Figure 7). Several different mutations around the RGD motif were seen in strains with different genotypes and temperature sensitivities. The temperature-insensitive strain had amino acid substitutions at sites T283S, V284M, and R288K in the VP1 region, and the temperature-sensitive strain had the same amino acids translated at these three sites. We speculate that the amino acid substitutions at these three sites may be related to the temperature tolerance of the virus, but further animal experiments are needed to verify this.

## 4. Discussion

Previous studies have shown that more than 90% of EV meningitis occurs in association with EV-B [24]. The results of the National Enterovirus Surveillance System (NESS) in the United States from 1970–2005 showed that CVA9 was the seventh most common serotype, mainly causing neurological diseases such as AM and encephalitis, and causing 1.2% of case deaths [21]. Therefore, we should strengthen the surveillance of diseases associated with CVA9 to reduce its possible disease burden. In this study, we divided all CVA9 strains worldwide into 10 genotypes based on the criteria of 15–25% for defining genotype by nucleotide differences in enteroviruses [28]. The results of the genotyping were different from the previously reported genotyping results based on VP1/2A [5], 1D/VP1 [36], and partial VP1 [22]. The strains in G genotype are the currently dominant strains, which are mostly Chinese strains, but some are from Australia, the USA, Thailand, Poland, Canada, etc. This genotype has been diverging and spreading to other regions since 2005, causing the co-circulation of multiple transmission chains.

Recombinant analysis of enterovirus is crucial in epidemiology studies of the evolution of a virus and studies of new strains/types. Recombination analysis in this study revealed eight recombination patterns in China at present, CVB4, CVB5, E30, CVB2, E11, HEV106, HEV85, and HEV75. E14, and E6 may act as recombinant donors in multiple regions. Temperature resistance could be due to mutation in the genome or recombination, but the comparation of different temperature-sensitive strains showed that there was no significant difference in breakpoint positions except the donor sequences. One study performed in Canada speculated that one CVA9 strain isolated in 2010 arose from recombination events between one CVA9 strain isolated in 2003, CVB6, EV-86, and EV-100, and the finding was examined by SimPlot analysis [23]. Complete genome sequence analysis of two CVA9 strains isolated in Yunnan in 2009 demonstrated that recombination sites were probably located in the 2C–3C region, 3C–3D region, and 3D region, which were most closely related to E25, CVB1, and E30 [15]. CVA9 are highly homologous from the 5′UTR to the 2B domain. Recombination with HEV85 and E6 in the 5′ UTR clearly occurred beyond this point, which suggest that the increase in CVA9 cases likely did not result from the emergence of a radically different immune escape [23]. Further virological studies, such as reverse genetic studies, are needed to understand the role of genetic recombination in CVA9 evolution.

The analysis presented in this study suggests that CVA9 exhibits continuous migration, taking place over multiple decades and across vast geographical areas. Phylogeographic analysis indicated five significant migration pathways, four of which were from the USA to the UK, Tunisia, China, and Thailand, respectively, and another was from France to Romania (decisive support with BF ≥1000). Very strongly and strongly supported diffusion rates were also indicated between numerous regions worldwide. The phylogeographic patterns may have been influenced by missing data. However, the virus’s migration could also be tracked in previous studies about spatial dynamic analysis of CVA9. For example, one study performed by Hietanen found CVA9 isolates from the Asia–Pacific region collected between 2006 and 2013, US isolates collected in 2015 and 2016, and Finnish isolates collected in the 2000s exhibited possible relatedness to sequences collected from the US between 1970 and 1980 [36]. Additionally, another study also provided evidence for the migration patterns of CVA9 between different European countries and Tunisia [34]. In conclusion, understanding possible migration events between isolates originating from different geographical areas will be important in understanding how the species has evolved and what effects it might have on epidemiology.

The temperature sensitivity test is a good indicator of virulence and contagiousness [35]. A comparison of temperature sensitivity within different genotypes revealed that a high-temperature environment has an inhibitory effect on the replication of the CVA9 of G genotype. Although the G genotype is the dominant genotype of CVA9, which includes mostly strains from nine countries in Asia, North America, Oceania, and Europe, the resistance to temperature changes of strains in China_XJ_2018_99 (genotype I) showed that the strains in genotype I may have high virulence and potential contagiousness. The temperature-resistant strain had amino acid substitutions at sites T283S, V284M, and R288K in the VP1 region, and the temperature-sensitive strain had different amino acids translated at these three sites. We speculate that the amino acid substitutions at these three sites may be related to the temperature tolerance of the virus, but further animal experiments are needed to verify this. The whole study was based on the availability of the isolates related to the symptomatic diseases, which could have biased the results. Viral development is of concern, given the widespread circulation of non-polio enteroviruses, and especially given the role of CVA9 as the most common cause of meningitis worldwide, and their involvement in other severe neurological conditions, such as acute flaccid myelitis and encephalitis [37]. We should strengthen the monitoring of enteroviruses and use a variety of emerging technologies to improve the detection rate in samples in order to identify those that may cause outbreaks as early as possible.

## Figures and Tables

**Figure 1 viruses-14-00822-f001:**
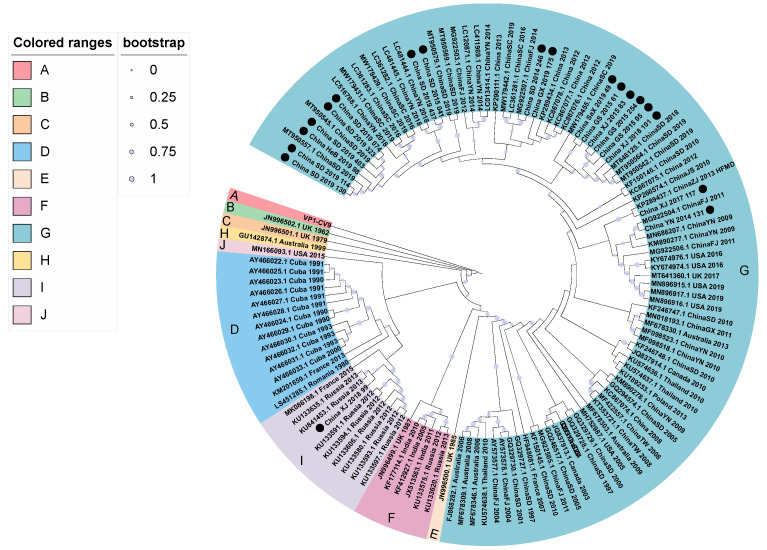
Phylogenetic tree based on complete VP1 nucleotide sequences of coxsackievirus A9 (CVA9) strains. ● indicates the strains in this study.

**Figure 2 viruses-14-00822-f002:**
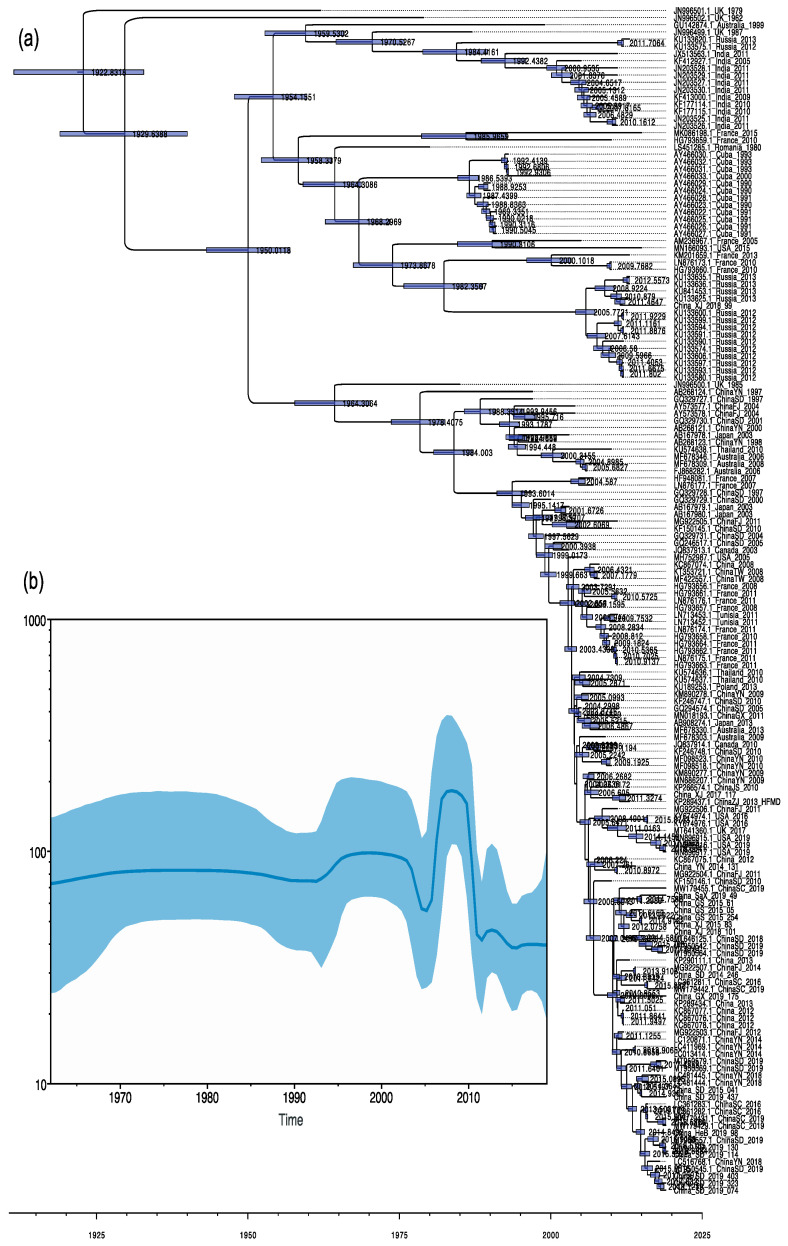
(**a**) Time-scaled phylogenetic tree generated using the MCMC method for 165 partial VP1 CVA9 sequences from the worldwide. Bars at notes indicate 95% HPDs of tMRCAs. The tree was node-labeled with inferred dates of linage splits. (**b**) Bayesian skyline plot of the CVA9 strains.

**Figure 3 viruses-14-00822-f003:**
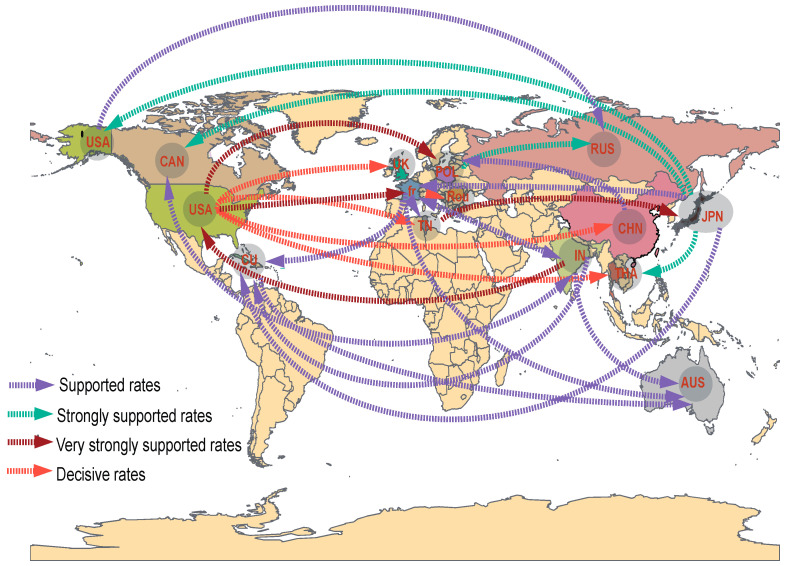
Spatial diffusion pathways of state transition for CVA9. Epidemiology unidirectional pathways from one location to another are indicated on the maps. Shown are only the state transitions with supported BF ≥ 3. Orange arrows, decisive rates with BF ≥ 1000; red arrows, very strongly supported rates with 100 ≤ BF < 1000; green arrows, strongly supported rates with 10 ≤ BF < 100; purple arrows, supported rates with 3 ≤ BF < 10.

**Figure 4 viruses-14-00822-f004:**
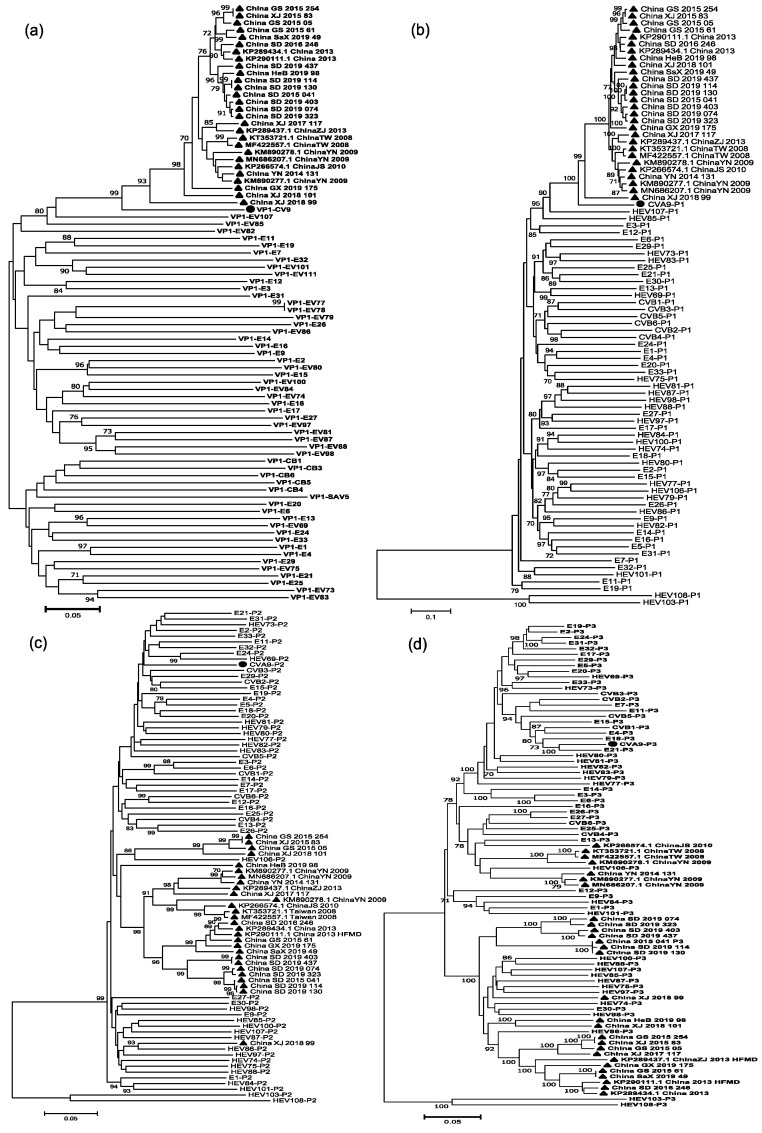
Neighbor-joining phylogenetic trees based on VP1, P1, P2, and P3 regions of the prototype sequence of all EV-B in the GenBank database and sequences of CVA9 strains in China. Numbers on codes indicate the bootstrap support of the node (1000 bootstrap replicate percentage). Scale bars represent the replacement of each site per year. Coding sequences of (**a**) VP1, (**b**) P1, (**c**) P2, and (**d**) P3 are shown. ● indicates CVA9 prototype strain (Griggs); ▲ indicates the Chinese CVA9 strains.

**Figure 5 viruses-14-00822-f005:**
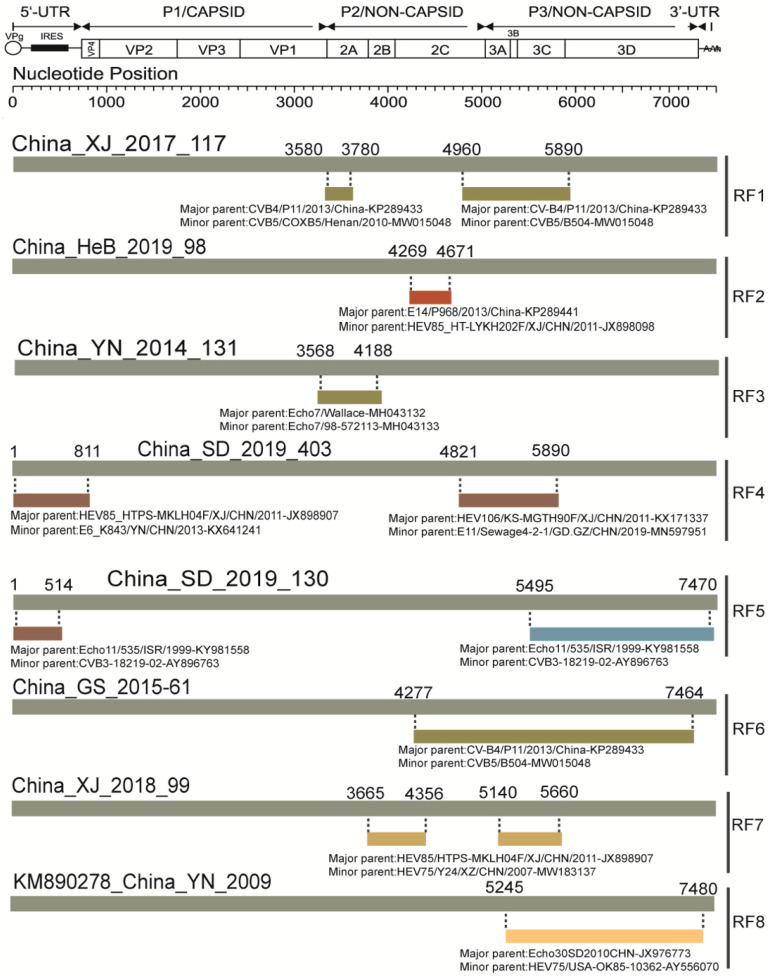
The genomic map of CVA9 representative strains recombination events predicted by RDP4. The grey band represents the full-length genome of the CVA9 strains; the numbers above indicate beginning and ending breakpoint positions. The different-colored bands represent the genomic regions where recombination events may have occurred; the numbers below indicate major and minor parents of the predicated recombination event.

**Figure 6 viruses-14-00822-f006:**
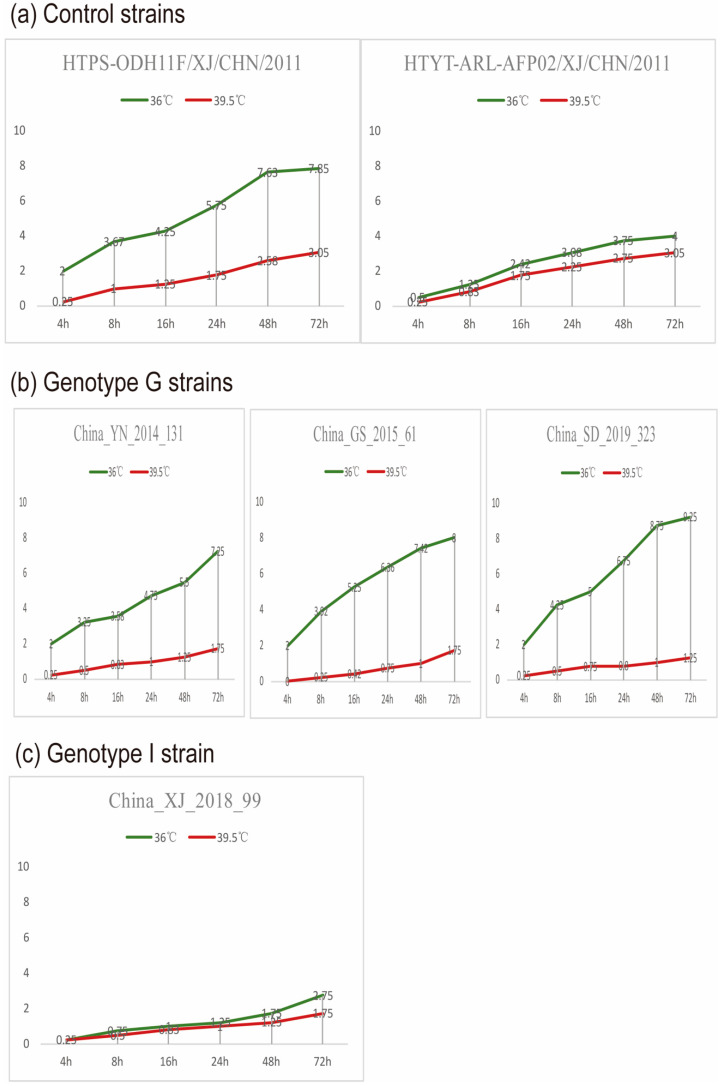
A titer timeline chart of four CVA9 temperature sensitivity experiments. The green and red lines represent line charts of the virus titer changing over time at 36 °C and 39.5 °C, respectively. Xinjiang strain EV-B106 (HTPS-ODH11F/XJ/CHN/2011) and Xinjiang strain EV-B85 (HTYT-ARL-AFP02F/XJ/CHN/2011) were used as temperature-sensitive and temperature-insensitive controls, respectively. (**a**) temperature sensitivity of control strains; (**b**) temperature sensitivity of representative strains in genotype G; (**c**) temperature sensitivity of representative strains in genotype I.

**Figure 7 viruses-14-00822-f007:**
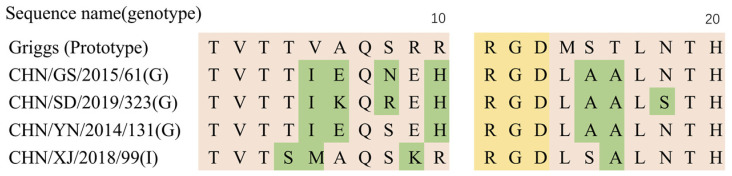
CVA9 RGD binding site analysis showing the fully conserved nature of the RGD motif in the four representative strains. VP1 position 290 in the protype strain Griggs is equivalent to position 11 in the alignment.

**Table 1 viruses-14-00822-t001:** The characteristics of eight recombination forms of Chinese CVA9 and their representative strains.

Pattern	Region	Name of Strains or GenBank Accession	Representative Strain
No recombination	N/A	MF42255, KT352721, GX19–175	N/A
RF1	P2, P3	KP289434, KP290111, KP289437, SD/2016/246, XJ/2017/117, SaX/2019/49	XJ/2017/117
RF2	P2	HeB/2019/98, XJ/2018/101	HeB/2019/98
RF3	P2	YN/2014/131, KP266574, MN686207, KM890277	YN/2014/131
RF4	5′, P2	SD/2019/074, SD/2019/323, SD/2019/403, SD/2019/437, SD/2015/041	SD/2019/403
RF5	5′, P2, P3	SD/2019/114, SD/2019/130	SD/2019/130
RF6	P2, P3	XJ/2015/83, GS/2015/05, GS/2015/61, GS/2015/254	GS/2015/61
RF7	P2, P3	XJ/2018/99	XJ/2018/99
RF8	P3	KM890278	KM890278

## Data Availability

Not applicable.

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
