# Peer review of "Molecular Epidemiology and Evolution of Coxsackievirus A9"

_viruses, 2022, doi:10.3390/v14040822_

Round 1
Reviewer 1 Report
In the present manuscript Zhao and collaborators describe an interesting and complete study regarding to the molecular epidemiology of CVA9, and important etiology agent mainly of meningoencefalitis worldwide. The study could be consider for publicaction after some points could be enhanced.
Abtract
It will be more informative if the authors add:
The number of worldwide sequences that were included in the phylogenetic and phylogeographic inferences.
The number of the sequences of the recombination analysis and respect to whom it was done, in this case: ( number of the references sequences) whithin the enterovirus B specie.
Genotipo A it is not mentioned.
Line 29: It will be more clear if the authors especified that the TMRCA metioned correspon to the reference strain or to this serotype.
Introduction
Line 68-69 Insertar respectivelly, referring to percents of nucleotide differences
Insert references to the ideas in:
Line 98: after worlwide.
Line 101: transmission of CVA9.
Line 106: BEAST program.
Material and Methodos
2.1. Virus isolation
Line 116: It is not clear the first idea, it sound that the 19 isolates where reactivated on RD cells, however, by the results it seems clinical samples but were described as isolates.
2.2. Datasets construction of worldwide and Chinese CVA9
Could the authors include the specific coding regions for each analysis, for example: complete VP1 for phylogeny, P1 for phylogeography, and complete sequences for recombination.
As well as, I consider that it will be more descriptive if the authors explain for the two first analysis:
The assamble of the sequences to obtain the complete sequence (done, change for this section line 138-140).
The aligment of wich sequences (e.g. all VP1 or CVA9 complete sequences available at GenBank) and the respective program used for that.
2.3. Phylogenetic analysis
In my opinion, this section could be rewrite to by more clear. The authors describe a phylogeny by NJ as method and Kimura 2-parameter substitution model with 1000 bootstrap replicates using MEGA (version 7.0) software, and then they mentioned also the GTR+G+I model evaluated by Modeltest (version 3.7) software. Maybe, they could describe firstly the evolutionary model evaluated for the specific data; then the evaluation by TempEst (version 1.5.3); and finally the phylogeny constuction.
I wonder, why the authors don't used the same data and the BEAST interface for the phylogenetic and the phylogeographic analysis?. Taking into account that as showed in the first part of recombination study, complete VP1 And P1 phylogeny were equivalent in terms of recombination signal, VP1 set of sequences include more sequences that the data set for phylogeography and the roboustness of the BEAST program can obtain both analysis simultaneously base on the same data set.
Reference all the bioinformatic programs used (BEAST, Tracer , etc.)
2.4. Recombination analysis
In my opinion this section it could be after 2.5. Discrete phylogeography analyses.
In this section the authors do not describe the methodology used to obtain the four phylogenies (VP1, P1, P2, P3) that are showing in Results section (Figure 2).
2.5. Discrete phylogeographic analysis
In this section, in my opinion, the authors could rename it because they not only obtain the phylogeographic analysis, also the rest of the information, e.i., Skyline analysis, TMRCA, the nucleotide sustitution rate, know all of those as Filodinamic analysis. (Grenfell BT PO, Gog JR, Wood JL, Daly JM, Mumford JA, et al. Unifying the epidemiological and evolutionary dynamics of pathogens. Science. 2004;303(5656):327-32; Volz EM, Koelle K, Bedford T. Viral phylodynamics. PLoS Comput Biol. 2013;9(3):e1002947).
I consider that the authors could include all the information of the set of sequences selected for that analysis, genomic region, alignment method etc. in case that they dont include this information in at 2.2 Data set.. section as recomended before.
The authors also could include the program to visualize the tree.
Line 158: change phytogeography by phylogeographic
Resultados
3.2. Dataset description
Could be more complete if the authors include all the clinical and epidemiological data of the patients from who the 19 strains were collected, since all the strains were reported by HFMD surveillance system. Even that, this could be interesting beacuse several of the sequences in the analysis came from meningoencefalitis cases. The authors could discuses about the diferents clinical presentation of CVA9 since the genetic relationship obtained in the phylogeny.
Line 202: replace CA9, by CVA9.
Line 203: replace 1850s by 1950s, Griggs strain date from 1950 by the Table S2.
Line 204: replace America by USA.
3.3. Phylogenetic analysis
It would be more complete if the authors include a summary comment regarding the evaluation of the data by TempEst and the output of the program in the supplementary data.
Following the order suggesting in section 2.3 the authors could rewrite the present section: genomic region (nt), number of the sequences, evaluation of the data, and the program used for the analysis if MEGA or BEAST.
Line 220: Cuban strains where isolated between 1990-1993, and 2000.
Is denoted in the phylogeny the Venezuela strain 2018 (MK652143. 1) placed alone in a branch just after reference strain in both trees, Don't think the authors that it could be a possible recombinant whose parents could be CVA9 strains?. Recombination analysis is based on the data that is selected, and in this case this phenomeno were studied only in the data set composed by the 19 Chinese sequences and the rest of the reference enterovirus B strains and not whithin CVA9 sequences data set.
Line 235: Figure 1. Could include the number of sequences and the lenght of VP1 region as well the program and evlutionary model. It will be more informative if the authors place the bootstrap values at least on the Branch or node that define the different genotypes, as well as they could higlight in any way the 19 chineses strains of this study.
3.4. Analysis of recombination patterns of CVA9
As mentioned before this section it could be after 3.5. Discrete phylogeographic analyses.
It could be more informative if the authors describe the number of the sequences that were included for the phylogenetic analysis of the VP1, P1, P2, and P3, as well models and program used for that.
Line 249-265: In my opinion this section could be rewriter in order to improve the description as follow: first to mention the strain name of the recombinant, genotype, since the main objective is to identified recombinants in the 19 sequences obtained, then thier respective parents, and less importan in cronological order.
Please check the Chinese sequence China_XJ_2018_99 it was isolated in 2015 or 2018.
Line 267: Figure 2. The authors could describe what means the filled triangle and filled circle.
3.5. Discrete phylogeography analyses
In my opinion as mentioned before this section could be remane as Phylodinamic analyses.
In the time-scaled phylogenetic tree (Figure 4), it is also denote the influence of the Venezuela strain 2018 (MK652143). In this tree has an comon ancestor with the ancientest strain (JN996502 dated on 1962), where the time for this ancestor show 1905.1799 as date, however the nearest node that include the two mentioned strains and JN996501 strain (dated on 1979) show 1879.8096 as date. How the author could explain this results?, they could spect that this date should be reasonably after 1905.1799 and not 26 year before, if it is include a younger strain?. Could be difficult to the authors try to look for recombiation events in a CVA9 set of sequences where this Venenzuelan strains is include because is not published the complete sequence and the most similar sequences have around 80 % of identity by BLAST, they could try. They could think to another explanation as any mistake at lab, with the greatest respect to the colleagues who published it, but in my opinion is not sence reasonable the homology of this strain obtained in 2018 with the most ancestrals strains of the data set, even when the behavior of other sequences of the same year although from different countries is not the same.
Line 293: Figure 4. The Bayesian skyline analysis is not mentioned. Maybe one value after coma is fine and informative to the time at the node values. Bars at nodes indicate 95% HPDs of TMRCAs.
Line 298: change regions by countries and America by USA. Line 311:
Figure 6: Only are shown the state transition with supported BF (>=3)
Line 315: Delete xiao yu deng
3.6. Temperature sensitivity properties of the CVA9
It would be more informative if the authors provided a simple explanation of the purpose of these study just to introduce it.
The term Figure 7 is not mentioned in the text.
Figure 7: In my opinion, the figure could be improve, for example, in any way higlight, organice in the space or differentiate the controls graphic, as well as by genotipe, maybe identify with numbers or leters: Controls strains (A), genotype F (B), genotype G strains (C).
Suplementary materials
It will be more informative if the authors:
Include the description of each Table.
Table S1: Insert a colum to describe the specific genomic region that is amplified by each primer.
Table S2: Organize in any way maybe by cronologically and/or genotype (if it is include in a new colum).
Table S3: Check some strains are doble, e.x., 16C2 (KY674974) and 16C8 (KY674976). Please change America by USA, this table could be organized also maybe cronologically.
Table S4: Maybe it is more informative if only are shown the rutes with BF>= 3.
Author Response
Response to Reviewer 1 Comments
Abstract
It will be more informative if the authors add:
Point1: The number of worldwide sequences that were included in the phylogenetic and phylogeographic inferences.
Response 1: We are all in agreement with your advice and added the number of worldwide sequences that were included in the phylogenetic and phylogeographic inferences. Phylogenetic analysis is based on the complete VP1 of CVA9 all over the world, and there are 129 sequences available to do this analysis (combined with the 19 sequences in this study and 35 complete CV9 sequences in the GenBank). Because of the CVA9 sequence isolated in Venezuela in 2018(MK652143. 1) placed alone in a branch just after reference strain in phylogenetic tree (as shown in the figure 1 on the right), we agree with the guess that it could be a possible recombinant whose parents could be CVA9 strains, and it may be the sequences whose genetic divergence and sampling date are incongruent (The dot in the upper right corner of Figure 1 is the Venezuelan strain). So, in the final analysis of the phylogenetic evolutionary, we deleted the Venezuelan strain, included 128 full-length VP1sequences for phylogenetic analysis.
Figure 1: The results of TempEst of the complete 129 VP1 sequences
For phylogeographic analysis, we downloaded all CVA9 sequences from GenBank whose length is 600-7500 bp, plus 19 sequences from this study, for a total set of 241 sequences of the VP1 region. Sequences in the non-VP1 region and short fragment were removed through the Sequencher software analysis, then leaving 166 sequences(723bp). These 166 sequences were incorporated into MEGA software for alignment, and NJ tree were constructed and exported to NWK format tree files, which were later imported into TempEst software to analyze whether there was sufficient temporal signal in the selected sequences for phylogenetic molecular clock analysis. Delete the Venezuela strain isolated in 2018, there were 165 partial VP1 strains for phylogeographic analysis.
In total, 128 complete VP1 sequences of CVA9(906bp) were used for phylogenetic analysis and 165 partial VP1 sequences(723bp) were used for phylogeographic analysis. These data have been added in the abstract.
Point 2: The number of the sequences of the recombination analysis and respect to whom it was done, in this case: (number of the references sequences) within the enterovirus B specie.
Response 2: Thanks for your advice, there are 68 references sequences within the enterovirus B specie was used for the recombination analysis Supplementary table). The phylogenetic trees constructed between the Chinese CVA9 strains and other EV-B prototype strains based on the P2 and P3 region show the Chinese CVA9 strains are distant from the CVA9 prototype strain in both the P2 and P3 regions, suggesting that these Chinese CVA9 isolates may have recombination in the non-structural protein coding region. 68 EV-B prototype strains and 19 strains in this study were alignment using MEGA (version 7), and the output result was analyzed by RDP4 to find potential recombination and possible patterns of recombination, a representative strain of each recombination pattern was then selected for subsequent recombination analysis. To further investigate the source of CVA9 recombination, we performed BLAST comparison of the non-structural protein coding region sequences P2 and P3 regions of representative strains of CVA9 with different recombination patterns that were most likely to undergo recombination on the NCBI website, and selected the other EV-B with the highest nucleotide sequence similarity (90%) prevalent in China as the reference sequence for recombination analysis, and the EV-B of the same serotype with essentially the same similarity we selected the earlier year, and the same serotype was not repeatedly selected.
Point 3: Genotype A it is not mentioned.
Response 3: Thanks for your kind advice. We have added the description about Genotype A in the Abstract. The detail content is as follows: “The prototype strain was included in genotype A”.
Point 4: Line 29: It will be clearer if the authors specified that the TMRCA motioned correspond to the reference strain or to this serotype.
Response 4: Thanks for your advice. We have changed to the description in line 29 to “The mean substitution rate of partial VP1 region(723nt) of CVA9 was 3.27 × 10-3 substitutions/bit/year (95% HPD range 2.93-3.60 × 10-3) with a tMRCA dating back to 1922 (1911-1932). The spatiotemporal dynamics of CVA9 showed the spread of CVA9 had obviously increased in recent years”.
Introduction
Point 1: Line 68-69 Insertar respectivelly, referring to percents of nucleotide differences
Response 1: Thanks for your advice, and we have changed the sentence to “The proposal made by Oberste et al. to classify enterovirus genotypes and sub-genotypes according to 15-25% and 8-15% nucleotide differences respectively in the complete VP1 region followed combined with phylogenetic analysis has become the prevailing standard”, added the “respectively “in the sentence according to you, and it is exactly much clearer to complain the principle of the classification of the enterovirus genotype and sub-genotypes.
Point 2: Insert references to the ideas in (1) Line 98: after wordlwide. (2) Line 101: transmission of CVA9. (3) Line 106: BEAST program.
Response 2: Thanks for your advice. We have added references in the appropriate places based on your suggestions.
(1) Tao, Z.; Wang, H.; Li, Y.; Liu, G.; Xu, A.; Lin, X.; Song, L.; Ji, F.; Wang, S.; Cui, N.; Song, Y., Molecular epidemiology of human enterovirus associated with aseptic meningitis in Shandong Province, China, 2006-2012. PLoS One 2014, 9, (2), e89766.
(2) Chen, P.; Wang, H.; Tao, Z.; Xu, A.; Lin, X.; Zhou, N.; Wang, P.; Wang, Q., Multiple transmission chains of coxsackievirus A4 co-circulating in China and neighboring countries in recent years: Phylogenetic and spatiotemporal analyses based on virological surveillance. Mol Phylogenet Evol 2018, 118, 23-31.
(3)Suchard, M. A.; Lemey, P.; Baele, G.; Ayres, D. L.; Drummond, A. J.; Rambaut, A., Bayesian phylogenetic and phylodynamic data integration using BEAST 1.10. Virus Evol 2018, 4, (1), vey016.
Material and Methods
2.1. Virus isolation
Point 1: Line 116: It is not clear the first idea, it sounds that the 19 isolates where reactivated on RD cells, however, by the results it seems clinical samples but were described as isolates.
Response 1: We appreciate your question about the source of the samples. The strains in this study were all isolated from the HFMD surveillance system, they are indeed isolated strains. And we have revised the statement in the results about the samples. Since the establishment of the HFMD case samples surveillance system in China in 2008, HFMD surveillance has been carried out continuously for many years, covering 31 provinces in mainland China. The operational process is as follows: clinical samples from HFMD patients are collected by the laboratories of provincial CDC according to the clinical sample collection specifications in the Polio Laboratory Manual issued by the WHO. Sample types mainly include pharyngeal swabs, anal swabs or fecal specimens. After the collection is completed, the samples with the HFMD submission form will be sent to the Chinese Center for Disease Control and Prevention (CCDC) by the provincial CDC laboratory. After receiving the clinical samples, the CCDC laboratory will perform EV serotyping and genotyping, etc. Then the viral fluids isolated from above clinical samples are stored in the HFMD surveillance system. 19 CVA9 samples were from the HFMD surveillance system, we used the processed viral fluid to inoculate RD cells. Cells were continuously observed after inoculation for seven days, and the culture was harvested if cells showed a complete EV-like cytopathic effect.
2.2. Datasets construction of worldwide and Chinese CVA9
Point 1: Could the authors include the specific coding regions for each analysis, for example: complete VP1 for phylogeny, P1 for phytogeography, and complete sequences for recombination.
Response 1: Thanks for your advice, and we have changed the description about the datasets. Changed the description to “Complete VP1(906bp) for phylogenetic analysis, partial VP1(723bp) for phylogeographic analysis, and 19 complete sequences in this study were used for analysis of recombination”.
As well as I consider that it will be more descriptive if the authors explain for the two first analysis:
Point 2: The assemble of the sequences to obtain the complete sequence (done, change for this section line 138-140).
Response 2: Thanks for your advice, and we have changed the description. Sequencing data were stored as standard chromatogram format files and assembled based on prototype by using Sequencher software (version 5.4.5). Sequences of 906-7500 bp in GenBank were recruited and cropped according to the full-length VP1 sequence of the prototype strain using Sequencher software, removed Delete non-VP1 region sequences and sequences that do not match the length of the full-length VP1 region. Then 129 full-length VP1 region sequences were selected and exported.
Point 3: The aligment of wich sequences (e.g., all VP1 or CVA9 complete sequences available at GenBank) and the respective program used for that.
Response 3: Before phylogenetic analysis, we performed sequence screening and shearing using Sequencher software to remove sequences in the non-VP1 region and sequences that do not match the length of 906nt, and then a total of 129 complete VP1 sequences were screened, while identifying the suspicious Venezuelan strain, so we removed it, a total of 128 sequences were selected for phylogenetic analysis. We compared 128 full-length VP1 region sequences and constructed a maximum likelihood evolutionary tree based on the screened evolutionary model of GTR+G+I in MEGA.
For the phylogeographic analysis, we also used Sequencher software to screen and shear the sequences in this study and sequences of 600-7500nt in GenBank, and selected a dataset representing the widest geographic range and time span for phylogeographic analysis, and later selected a total of 166 sequences by comparison analysis in Sequencher. While identifying the Venezuelan strain whose genetic divergence and sampling date are incongruent, we also deleted it in this analysis, so a total of 165 partial VP1 sequences(723nt) were used for phylogeographic analysis. This dataset contains 14 countries worldwide and spans the time period 1950-2019.
Finally, 128 full-length VP1 region sequences (906nt) and 165 partial VP1 region sequences (723nt) were compared and analyzed for phylogenetic and phylogeographic analyses, respectively.
2.3. Phylogenetic analysis
Point 1: In my opinion, this section could be rewrite to by clearer. The authors describe a phylogeny by NJ as method and Kimura 2-parameter substitution model with 1000 bootstrap replicates using MEGA (version 7.0) software, and then they mentioned also the GTR+G+I model evaluated by Modeltest (version 3.7) software. Maybe, they could describe firstly the evolutionary model evaluated for the specific data; then the evaluation by TempEst (version 1.5.3); and finally, the phylogeny constuction.
Response 1: Thanks for your advice, and we have rewritten this paragraph. Firstly, we recruited 110 full-length VP1 sequences in GenBank (including the VP1 region sequences in the full-length part intercepted by Sequencher software) together with 19 CVA9 sequences in this study. The best evolutionary model was selected by the Modeltest(version 3.7) software, and the neighbor-joining tree was constructed based on the selected GTR+G+I evolutionary model. The tree file was exported to NWK format and imported into TempEst for data screening, and it was found that the CVA9 strain isolated from Venezuela in 2018 had an exceptionally long evolutionary branch between the prototype strain and the UK strain isolated in 1962, which was probably a sequencing error, or it is a sequence whose genetic divergence and sampling date are incongruent. So, the sequence was removed from the analysis. Then the ClaustW tool in MEGA software was used to a new alignment for the remaining sequences, and the maximum likelihood (ML) tree was constructed with the GTR+G+I evolutionary model using MEGA.
Point 2: I wonder, why the authors don't used the same data and the BEAST interface for the phylogenetic and the phylogeographic analysis? Taking into account that as showed in the first part of recombination study, complete VP1 And P1 phylogeny were equivalent in terms of recombination signal, VP1 set of sequences include more sequences that the data set for phylogeography and the roboustness of the BEAST program can obtain both analyses simultaneously base on the same data set.
Response 2:Thanks for your advice, we think your suggestion “VP1 set of sequences include more sequences that the data set for phylogeographic analysis and the robustness of the BEAST program can obtain both analyses simultaneously base on the same data set” is more descriptive to show the spatial-temporal dynamic transmission. We found that the differences in evolutionary rate and time of origin were not particularly large by comparing the phylogeographic analysis of partial VP1 sequences(723nt) and P1 region sequences(2607nt)(Table 1). In order to analyze the dynamic transmission pathways among more countries, we finally selected partial VP1 region sequences for phylogeographic analysis, adding sequences from Japan, Tunisia and Romania, and the maximum clade credibility (MCC) tree also confirms the correctness of the genotypic classification method in 2.3 Phylogenetic analysis.
Substitution Rate (10-3subs./site/year) * |
|
tMRCA(year)# |
||
VP1 |
P1 |
|
VP1 |
P1 |
3.27 (2.93-3.60) |
3.53 (2.54-4.31) |
|
1922 (1911-1932) |
1919 (1881-1941) |
Note: *Estimated mean substitution rates (95%HPD range in parentheses).
#Estimated year of tMRCA (95% CI in parentheses).
Point 3: Reference all the bioinformatic programs used (BEAST, Tracer, etc.)
Response 3: Thanks for your kind suggestions, and we have added references in the appropriate places based on your suggestions.
BEAST: Suchard, M. A.; Lemey, P.; Baele, G.; Ayres, D. L.; Drummond, A. J.; Rambaut, A., Bayesian phylogenetic and phylodynamic data integration using BEAST 1.10. Virus Evol 2018, 4, (1), vey016.
Tracer:Rambaut A, Drummond AJ, Xie D, Baele G, Suchard MA. Posterior Summarization in Bayesian Phylogenetics Using Tracer 1.7. Syst Biol. 2018 Sep 1;67(5):901-904. doi: 10.1093/sysbio/syy032.
Figtree: http://tree.bio.ed.ac.uk/software/figtree/
2.4. Recombination analysis
Point 1: In my opinion this section it could be after 2.5. Discrete phylogeography analyses.
Response 1: Thanks for your advice, we have put this section (Recombination analysis) to the 2.5 Discrete phytogeography analyses (the title of 2.5 has been changed to “Phylodynamic analysis”) according to you.
Point 2:In this section the authors do not describe the methodology used to obtain the four phylogenies (VP1, P1, P2, P3) that are showing in Results section (Figure 2).
Response 2:Thanks for your suggestion about the methodology used to obtain the four phylogenies (VP1, P1, P2, P3)and we have added the methodology to the 2.5. Recombination analysis. For the modification of this paragraph, we placed the recombination analysis after the phylogeographic analysis. And we added the methodology used to obtain the four phylogenies (VP1, P1, P2, P3). The description as “For the recombination analysis of Chinese strains, we selected 68 sequences of EV-B prototype strains in the GenBank, 9 full-length Chinese CVA9 sequences from GenBank and 19 sequences from this study to construct the neighbor-joining (NJ) phylogenetic trees of VP1, P1, P2 and P3 coding regions. The sequences were processed by MEGA with the best evolutionary model (GTR+G+I), and the NJ tree was constructed in MEGA (version 7) with 1000 bootstrap replicates”.
2.5. Discrete phylogeographic analysis
Point 1: In this section, in my opinion, the authors could rename it because they not only obtain the phylogeographic analysis, also the rest of the information, e.i., Skyline analysis, TMRCA, the nucleotide sustitution rate, know all of those as Phylodynamic analysis. (Grenfell BT PO, Gog JR, Wood JL, Daly JM, Mumford JA, et al. Unifying the epidemiological and evolutionary dynamics of pathogens. Science. 2004;303(5656):327-32; Volz EM, Koelle K, Bedford T. Viral phylodynamics. PLoS Comput Biol. 2013;9(3):e1002947).
Response 1: Thank you very much for your professional advice. We have carefully read your recommended literature and found that using your recommended " Phylodynamic analysis " is more suitable. And we have changed the “2.5. Discrete phylogeographic analysis” to “2.5. Phylodynamic analysis”.
Point 2: I consider that the authors could include all the information of the set of sequences selected for that analysis, genomic region, alignment method etc. in case that they don’t include this information in at 2.2 Data set. section as recommended before.
Response 2: Thanks for your kind suggestion, and we added the information of the set of sequences selected for this analysis in 2.2 Data set. And we also added the description in this part. 165 partial VP1 sequences(723nt) were selected for phylogeographic analysis. This dataset contains 14 countries worldwide and spans the time period 1950-2019. Sequence alignment was conducted using the ClustalW tool in MEGA (version 7.0), the global evolutionary dynamic of CVA9 over time were inferred based on the partial VP1 capsid region. The correlation coefficient and regression value of each dataset were calculated using TempEst (version 1.5.1) to estimate the correlation between sequence divergence and the date of isolation in each dataset. The Markov Chain Monte Carlo (MCMC) method implemented in BEAST (version 1.10.4) was used to estimate the temporal phylogenies and rate of evolution. All 165 partial VP1 sequences were analyzed using the strict clock and constant site tree prior with the GTR+G+I nucleotide substitution model. A Bayesian MCMC run consisted of 200000000 generations to ensure that each parameter could converge. The sampling of frequency was set to 20000 generations. The output from BEAST was analyzed using TRACER (version 1.7.1) (with estimated sample size (ESS) values higher than 200). A maximum clade credibility (MCC) tree was constructed using TreeAnnotator, with the burn-in option used to remove 10 percent of sampled trees, and the resulting tree was visualized by FigTree (v1.4.4).
Point 3:The authors also could include the program to visualize the tree.
Response 3:Thanks for your kind advice. The obtained trees were visualized with the FigTree (version 1.4.4) software, and we have added the program used in the article.
Point 4: Line 158: change phytogeography by phylogeographic
Response 4: Thanks for your advice, we have changed “phytogeography” to “phylogeographic” in Line 158.
Resultados
3.2. Dataset description
Point 1:Could be more complete if the authors include all the clinical and epidemiological data of the patients from who the 19 strains were collected, since all the strains were reported by HFMD surveillance system. Even that, this could be interesting because several of the sequences in the analysis came from meningoencephalitis cases. The authors could discuss about the different clinical presentation of CVA9 since the genetic relationship obtained in the phylogeny.
Response 1: Thanks for your advice, 19 strains in this study were obtained from Shandong (8), Gansu (3), Xinjiang (4), Shaanxi (1), Yunnan (1), Hebei (1) and Guangxi (1) based on HFMD surveillance system, of which 3 strains were from severe HFMD cases in Gansu and the rest from mild cases. And we added above in the article.
We believe you said that it could be interesting cause several of the sequences in the analysis came from meningoencephalitis cases, and we added the discussion about the different clinical presentation of CVA9 in the discussion. The clinical symptoms of the AM patients were fever (>37.5°C), headache and vomiting. A majority of these patients were 4-5 years old. In contrast, patients with HFMD have a rash or herpes on the hands, feet, and around the mouth, and some severe cases are accompanied by fever, and the rash and herpes often spread throughout the body, especially on the limbs, trunk, and neck. Although AM and HFMD have different clinical symptoms, the phylogenetic tree showed the strains in this study isolated from HFMD and those from aseptic meningitis (AM) did not differ significantly in genotype distribution, as most of them belonged to genotype G and were in a transmission chain.
Point 2: (1) Line 202: replace CA9, by CVA9. (2) Line 203: replace 1850s by 1950s, Griggs strain date from 1950 by the Table S2. (3) Line 204: replace America by USA.
Response 2: Thanks for your advice, we have changed the Line 2020, line 203 and line 204 by CVA9, 1950s and USA, respectively.
3.3. Phylogenetic analysis
Point 1: It would be more complete if the authors include a summary comment regarding the evaluation of the data by TempEst and the output of the program in the supplementary data.
Response 1: Thanks for your advice, we added a summary comment regarding the evaluation of the data by TempEst and the output of the program supplementary data.
Point 2: Following the order suggesting in section 2.3 the authors could rewrite the present section: genomic region (nt), number of the sequences, evaluation of the data, and the program used for the analysis if MEGA or BEAST.
Response 2:Thanks for your advice, and we rewrote this section following your suggestion in section 2.3. 128 complete VP1 sequences(906nt)were used for phylogenetic analysis. Sequence alignment was conducted using the ClustalW tool in MEGA (v7.0) and the best nucleotide substitution model was selected by “Find Best Model” tool in MEGA. The correlation coefficient and regression value of this dataset were calculated using TempEst to estimate the correlation between sequence divergence and the date of isolation in each dataset. Then the selected CVA9 were further divided into genotypes using maximum likelihood method (ML) in MEGA based on the best nucleotide substitution model.
Point 3: Line 220: Cuban strains where isolated between 1990-1993, and 2000.
Response 3: Thanks for your advice. We changed this sentence in Line 220 to “Cuban strains where isolated between 1990-1993, and 2000”.
Point 4: Is denoted in the phylogeny the Venezuela strain 2018 (MK652143. 1) placed alone in a branch just after reference strain in both trees, don’t think the authors that it could be a possible recombinant whose parents could be CVA9 strains? Recombination analysis is based on the data that is selected, and in this case this phenomeno were studied only in the data set composed by the 19 Chinese sequences and the rest of the reference enterovirus B strains and not whithin CVA9 sequences data set.
Response 4:Thanks for your advice, and we don’t think it is a recombinant whose parents could be CVA9 strains, but a sequence whose genetic divergence and sampling date are incongruent. Because we did BLAST using VP1 and P1 regions of the Venezuela strain 2018 (MK652143. 1), found that the percent identity with the CVA9 strains are more than 80% (D00627, JN996502, JN996501, KM201659, KU574638, AY466028 etc.) And we agree with your opinion about the description “in this case this phenomenon was studied only in the data set composed by the 19 Chinese sequences and the rest of the reference enterovirus B strains and not within CVA9 sequences data set.” We redid the restructuring analysis. First, we used 28 selected Chinese CVA9 sequences to construct phylogenetic trees with 59 prototype strains in EV-B for different segments (P1, VP1, P2, P3), with a view to discovering possible recombination positions of sequences and possible sequences with possible recombination in the EV-B group. Based on the 28 full-length Chinese CVA9 sequences, we used MEGA software to construct phylogenetic trees based on the nucleotide sequences of VP1, P1, P2 and P3 region sequences for 28 CVA9 whole genome representative sequences, as well as all EV-B prototype strains, and initially determined the possible recombination patterns by observing the different positions of each CVA9 strain on each evolutionary tree, with the formation of different clusters, and selected one representative strain for each recombination pattern for subsequent recombination analysis.In order to further investigate the recombination source of CVA9, we performed BLAST comparison of the non-structural protein coding segments of representative CVA9 strains with different patterns of optimal possible recombination on the NVBI website and selected other representative EV-B strains with the highest nucleotide similarity that are popular in China as reference sequences for recombination analysis. B of the same serotype with almost the same similarity, we selected the earlier ones, and the same serotype was not repeatedly selected, and finally the recombination analysis of CVA9 was performed based on the screened EV-B sequences.
Point 5: Line 235: Figure 1. Could include the number of sequences and the lenght of VP1 region as well the program and evlutionary model. It will be more informative if the authors place the bootstrap values at least on the Branch or node that define the different genotypes, as well as they could higlight in any way the 19 chineses strains of this study.
Response 5:Thanks for your advice. We have added the number of sequences and the length of VP1 region as well the program and evolutionary model in the Figure 1. And the bootstrap value added the branch of the phylogenetic tree. And the strains in this study were indicated by the ●
3.4. Analysis of recombination patterns of CVA9
Point 1: As mentioned before this section it could be after 3.5. Discrete phylogeographic analyses.
Response 1: Thanks for your advice, and this section has been placed to be after 3.5. Discrete phylogeographic analyses. And we changed the “3.5. Discrete phylogeographic analyses” to “2.5. Phylodynamic analysis”.
Point 2: It could be more informative if the authors describe the number of the sequences that were included for the phylogenetic analysis of the VP1, P1, P2, and P3, as well models and program used for that.
Response 2: Thanks for your kind advice. For the modification of methods of recombination analysis, we added the methodology used to obtain the four phylogenies (VP1, P1, P2, P3).
Point 3: Line 249-265: In my opinion this section could be rewriter in order to improve the description as follow: first to mention the strain name of the recombinant, genotype, since the main objective is to identified recombinants in the 19 sequences obtained, then thier respective parents, and less important in chronological order.
Response 3: Thanks for your advice, and We have modified this paragraph according to your suggestions. For the recombination analysis of Chinese strains, we selected 9 full-length sequences in GenBank and 19 sequences in this study, a total of 28 Chinese CVA9 strains for recombination analysis. Firstly, we used MEGA software to construct phylogenetic trees based on the nucleotide sequences of VP1, P1, P2 and P3 region sequences for 28 CVA9 whole genome representative sequences, as well as all EV-B prototype strains, and initially determined the possible recombination patterns by observing the different positions of each CVA9 strain on each evolutionary tree, with the formation of different clusters, and selected one representative strain for each recombination pattern for subsequent recombination analysis.
Table 2. The characteristics of eight recombination forms of Chinese CVA9 and their representative strain
Pattern |
Region |
Name of strains or GenBank accession |
Representative strain |
No recombination |
N/A |
MF42255、KT352721、GX19-175 |
N/A |
RF1 |
P2、P3 |
KP289434、KP290111、KP289437、SD/2016/246、XJ/2017/117、SaX/2019/49 |
XJ/2017/117 |
RF2 |
P2、P3 |
HeB/2019/98、XJ/2018/101 |
HeB/2019/98 |
RF3 |
|
YN/2014/131、KP266574、MN686207、KM890277 |
YN/2014/131 |
RF4 |
|
SD/2019/074、SD/2019/323、 SD/2019/403、SD/2019/437、SD/2015/041 |
SD/2019/074 |
RF5 |
|
SD/2019/114、SD/2019/130 |
SD/2019/114 |
RF6 |
|
XJ/2015/83、GS/2015/05、GS/2015/61、GS/2015/254、 |
GS/2015/05 |
RF7 |
5’, P2,3D |
XJ/2018/99 |
XJ/2018/99 |
RF8 |
|
KM890278 |
KM890278 |
Point 4: Please check the Chinese sequence China_XJ_2018_99 it was isolated in 2015 or 2018.
Response 4: Thanks for your advice. We have checked the Chinese sequence China_XJ_2018_99, it was isolated in 2018.
Point 5: Line 267: Figure 2. The authors could describe what means the filled triangle and filled circle.
Response 5: Thanks for your kind advice, we have added the means of the filled triangle and filled circle in the article. ●indicated other EV-B prototype strains; ▲indicated the CVA9 strains isolated in this study.
3.5. Discrete phylogeography analyses
Point 1: In my opinion as mentioned before this section could be remane as Phylodinamic analyses.
Response 1: Thanks for your advice. We have changed the “Discrete phylogeography analyses” to “Phylodynamic analyses” according to your advice.
Point 2: In the time-scaled phylogenetic tree (Figure 4), it is also denote the influence of the Venezuela strain 2018 (MK652143). In this tree has a comon ancestor with the ancientest strain (JN996502 dated on 1962), where the time for this ancestor show 1905.1799 as date, however the nearest node that include the two mentioned strains and JN996501 strain (dated on 1979) show 1879.8096 as date. How the author could explain this result? they could spect that this date should be reasonably after 1905.1799 and not 26 year before, if it is including a younger strain? Could be difficult to the authors try to look for recombination events in a CVA9 set of sequences where this Venezuelan strain is including because is not published the complete sequence and the most similar sequences have around 80 % of identity by BLAST, they could try. They could think to another explanation as any mistake at lab, with the greatest respect to the colleagues who published it, but in my opinion is not sence reasonable the homology of this strain obtained in 2018 with the most ancestral strains of the data set, even when the behavior of other sequences of the same year although from different countries is not the same.
Response 2:
We collected 222 sequences between 600-7500bp in GenBank, and analyzed them by Sequencher software, plus the 19 sequences in this study, a total of 241 sequences of VP1. The we put the 241 sequences into Sequencher software for comparison,removed non-VP1 sequences and short sequences(<723nt), leaving 166 partial VP1 sequences. The VP1 sequences with 723bp length were input into MEGA software for alignment analysis and output as Meg format file, and then "FIND BEST MODEL" program was used to filter the best evolutionary model, and the ML tree based on partial VP1 sequences was constructed according to the evolutionary model, and then the NWK format file was output, then the output NWK file was loaded into TempEst. The NWK file was loaded into TempEst software for data evaluation, and the Venezuelan strain was found to be slightly deviated, and based on the isolation time of 2018, it was considered that the strain might be a sequencing error strain, so after screening, we decided to delete the Venezuelan strain in this study. The final 165 partial VP1 sequences (723nt) remained for phylogeographic analysis, spanning 1950-2019 and covering a total of 14 countries worldwide.
Point 3: Line 293: Figure 4. The Bayesian skyline analysis is not mentioned. Maybe one value after coma is fine and informative to the time at the node values. Bars at nodes indicate 95% HPDs of TMRCAs.
Response 3: Thanks for your kind suggestion. We added the description about the Bayesian skyline analysis in the Figure 4 and also added the the note “Bars at nodes indicate 95% HPDs of TMRCAs”.
Point 4:Line 298: change regions by countries and America by USA.
Response 4: Thanks for your advice. We have changed regions by countries and America by USA.
Point 5: Line 311: Figure 6: Only are shown the state transition with supported BF (>=3)
Response 5: Thanks for your advice, we add “Only are shown the state transition with supported BF (>=3)” in the description of the figure 6.
Point 6: Line 315: Delete xiao yu deng
Response 6: Thanks for your advice, we have deleted “xiao yu deng”.
3.6. Temperature sensitivity properties of the CVA9
Point 1: It would be more informative if the authors provided a simple explanation of the purpose of these studies just to introduce it.
Response 1: Thanks for your advice, the temperature sensitivity test is a good indicator of virulence and contagiousness. (Sanders BP, de Los Rios Oakes I, van Hoek V, Bockstal V, Kamphuis T, Uil TG, Song Y, Cooper G, Crawt LE, Martín J, Zahn R, Lewis J, Wimmer E, Custers JH, Schuitemaker H, Cello J, Edo-Matas D. Cold-Adapted Viral Attenuation (CAVA): Highly Temperature Sensitive Polioviruses as Novel Vaccine Strains for a Next Generation Inactivated Poliovirus Vaccine. PLoS Pathog. 2016 Mar 31;12(3): e1005483. doi: 10.1371/journal.ppat.1005483.). And we added the mutations in RGD in results of 3.6. and the analysis of recombination in discussion in our article. Because previous studies have shown that the arginine-glycine-aspartic (RGD) motif found in the VP1 capsid protein of CVA9 has a role in cell entry. Analysis of amino acid variation in RGD fragments of different temperature-sensitive strains can provide a theoretical basis for studying the cellular receptors of CVA9, etc.
Point 2: The term Figure 7 is not mentioned in the text.
Response 2:Thanks for your advice, we added the figure 7 in the “Temperature sensitivity properties of the CVA9”.
Point 3:Figure 7: In my opinion, the figure could be improve, for example, in any way higlight, organice in the space or differentiate the controls graphic, as well as by genotipe, maybe identify with numbers or leters: Controls strains (A), genotype F (B), genotype G strains (C).
Response 3:Thanks for your advice, we have improved the figure 7 in the article.
Suplementary materials
It will be more informative if the authors: Include the description of each Table.
Point 1: Table S1: Insert a colum to describe the specific genomic region that is amplified by each primer.
Response 1: Thanks for your advice, we have inserted a column to describe the specific genomic region that is amplified by each primer according you.
Point 2:Table S2: Organize in any way maybe by cronologically and/or genotype (if it is include in a new colum).
Response 2: Thanks for your kind advice, we have sorted by genotype.
Point 3: Table S3: Check some strains are doble, e.x., 16C2 (KY674974) and 16C8 (KY674976). Please change America by USA, this table could be organized also maybe cronologically.
Response 3: Thanks for your advice, we have checked the table, there is no double sequences, and we have sorted by evolutionary time and genotype.
Point 4: Table S4: Maybe it is more informative if only are shown the rutes with BF>= 3.
Response 4:Thanks for your advice. We changed the information in Table S4. Only BF≥3 was shown.

Reviewer 2 Report
The manuscript “Molecular Epidemiology and Evolution of Coxsackievirus A9” has identified full sequences of 19 field isolates of CVA9 in different parts of China during 2010 and 2019. The manuscript reports the phylogenetic analysis, and the evolutionary clock of the virus isolates with attention to the possibility of the intertypic recombination.
There are some concerns regarding the manuscript which are presented point by point as appear in the manuscript.
The method section describes the assay for temperature sensitivity. It does not indicate that the virus quantification was performed in which conditions and why?
The result section shows the isolation of the strains in the RD cell line. Is there any possibility of recombination in cell culture environment? How did you rule it out?
You have discussed the genotypes A-H in section “3.3. Phylogenetic analysis”. How do you elucidate upon the reasons to these patterns of distribution of the genotypes? Are there any possible participating reasons? Is there any possibility that there is any pressure from the heard immunity?
Temperature resistance of the strains has been shown to exist, however, the relevance of the phenotype to the recombination or substitutions have not been discussed therefore it seems irrelevant to the whole manuscript. Although it has been discussed in the conclusion section the relevance of the temperature resistance to the pathogenicity of the virus, but it cannot be attributed to the specific genotype since the changes might be minor to render this resistance.
In general, the manuscript gives a good impression on the recombinant events in CAV9 and the migration of the infections between Australia and the USA and Canada. However, the issue of the evolutionary clock for the virus is based on the P1 region which might be the function of an immune pressure which can be variable during time and geographical regions based on epidemics history. Also, the issue of temperature resistance could be due to mutation in the genome or recombination which has not been discussed int the manuscript. Additionally, the whole study is based on the availability of the isolates related to the symptomatic diseases and the active surveillance to detect the virus in population simply might mean active case detection which could bias the whole study and might be good to be acknowledged.
Author Response
Response to Reviewer 2 Comments
The manuscript “Molecular Epidemiology and Evolution of Coxsackievirus A9” has identified full sequences of 19 field isolates of CVA9 in different parts of China during 2010 and 2019. The manuscript reports the phylogenetic analysis, and the evolutionary clock of the virus isolates with attention to the possibility of the intertypic recombination.
There are some concerns regarding the manuscript which are presented point by point as appear in the manuscript.
Point1: The method section describes the assay for temperature sensitivity. It does not indicate that the virus quantification was performed in which conditions and why?
Response 1: Thanks for your advice, four CVA9 strains were purified by plaque assay, and the titres of them were measured three times. The specific steps are as follows:
(1) Mix the purified virus isolate and dispense it into 10 tubes of the virus strain, about 80 µl per tube, and freeze it for storage;
(2) Add the suspension to a 96-well cell culture plate, 100μl per well, mark the date and incubate it in a CO2 incubator for titer determination the next day.
(3) Take 10 clean EP tubes of 1.5 ml, add 450μl of maintenance medium (MM) to each tube, then aspirate 50μl of the original virus solution mixed in the tube of the strain in the first tube (dilution 10-1), shake and mix thoroughly, then aspirate 50μl of the liquid in the first tube in the second tube (dilution of 10-2) and repeat the above steps until the dilution is 10-10 (Figure 2), the tip of the pipette gun needs to be changed during each dilution.
(4) Remove the 96-well culture plate with a single layer of RD cells in the incubator, pour off the growth medium (GM), add 100µl of MM to each well, and later add the diluted series of viral solution from low to high concentration to the 96-well cell culture plate sequentially (1-10 in total, 10 columns), making 4 replicate wells with 50µl of viral solution per well for each dilution gradient.
(5) Columns 11 and 12 of the 96-well cell culture plates as negative controls, seal the plates and mark them well, and place them in a 36°C CO2 incubator for static incubation.
(6) Observe cytopathic effect (CPE) in each well under an inverted microscope daily for 5~7 consecutive days and record the results.
(7) The results were judged positive when ≥50% of the cells in the wells showed CPE, and the viral titer of the titrated strain was calculated using the behrens-karber formula logTCID50=L-d(S-0.5).
(8) The experiment was repeated three times and the average value was taken as the final viral titer result to reduce experimental error.
And The method of virus quantification has been added in the method section.
Point 2: The result section shows the isolation of the strains in the RD cell line. Is there any possibility of recombination in cell culture environment? How did you rule it out?
Response 2: Thanks for your advice. We processed stool sample of HFMD patients according to standard procedures. Then the samples were inoculated into human rhabdomyosarcoma (RD) cells. The RD cells were supplied by American Center for Disease Control and Prevention. Cells were continuously observed after inoculation for seven days, and the culture was harvested if cells showed a complete EV-like cytopathic effect (CPE). The harvested culture was diluted with 4 gradients (10-2,10-3,10-4,10-5), and the diluted culture was inoculated into 6-well culture plates containing RD cells. Incubate at 35℃ for 2 hours, add agar medium without neutral red solution, and invert at 37℃ for 2 days, then add agar medium with neutral red solution, culture in the dark(35℃) after staining , and observed for seven days. After the appearance of white spots, appropriate plaques were selected and inoculated into RD cell culture tubes, after CPE appeared, they were passaged twice, and cultures were collected. Virus biological purification by the above virus plaque formation to isolate and purify CVA9 virus from the isolates. The purified strain was then used for virus isolation and subsequent sequence determination experiments. Because there is no donor sequence during cell culture, so it can be excluded that the recombination is in the process of cell culture.
Point 3: You have discussed the genotypes A-H in section “3.3. Phylogenetic analysis”. How do you elucidate upon the reasons to these patterns of distribution of the genotypes? Are there any possible participating reasons? Is there any possibility that there is any pressure from the heard immunity?
Response 3: Thanks for your suggestion. Based on the first reviewer's comments, we performed a new genotype classification of CVA9. We classified worldwide CVA9 strains into 10 genotypes A-J based on the 19 sequences in this study and sequences in GenBank using the full-length VP1 region sequence. The classification was based on the topology of the phylogenetic tree constructed from the full-length VP1 region and the evolutionary distance between nucleotides. 1999 Brown and Obsterte et al. first proposed criteria for the classification of genotypes and genotypes of EV-A71, and subsequently used 15-25% and 8-15% nucleotide sequence differences as the basis for the classification of enterovirus genotypes and genotypes. genotypes according to 15-25% and 8-15% nucleotide sequence differences. In addition to this approach, statistical methods for estimating the minimum evolutionary distance (P-distance) between groups (within groups) also yield the same results. Under the internationally accepted criteria for enterovirus typing, we combined the 19 sequences of the strains in this study with the CVA9 sequences available in GenBank for genotype classification and named them one by one in the chronological order of genotype discovery. From the phylogenetic tree we found that the distribution of these genotypes is clearly geographically clustered, so we strongly agree with your question about whether there is a population immunity effect on the distribution of genotypes.
We have the following conjectures about the reasons for the genotype distribution pattern:
(1) The epidemic intensity of the disease
Previous molecular genetic characterization for CVA9 was mainly based on studies of aseptic meningitis outbreaks, which tend to maintain a certain level of immunity in a population in a certain area for a certain period of time, and the timing of immunity after the disease, etc., also affects the spread of the virus and the distribution of genotypes.
(2) Forms of disease transmission
Most enteroviruses are mainly recessive infections, which makes the distribution of viruses mainly disseminated, and we can confirm this from the sequence distribution of CVA9 in GenBank. However, with the rapid development of economy and transportation, the frequency and speed of movement of people and materials will also enable the rapid movement of pathogens and infectious agents, thus causing the spread and epidemic of the virus worldwide.
(3) Nucleotide mutation and genetic recombination
The evolutionary mechanisms of the EV genome include nucleotide mutations and genetic recombination of genes. Although the genomes of RNA viruses are relatively short, their mutation rates are high due to the lack of RNA enzyme proofreading activity during genome replication. The mutation rate of viral structural proteins (P1 region) is higher than that of non-structural proteins (P2 and P3 regions) in a particular genome, which may be related to the host immune surveillance forcing the viral capsid proteins to undergo differentiation, etc. Therefore, nucleotide variation and gene recombination may also be a possible factor for the different distribution of genotypes.
(4) Herd immunity
The EV epidemic patterns at global and local geographic scales are mainly determined by herd immunity against each EV type. Other non-selective factors such as random transportation events of virus strains between countries by infected individuals may also be involved.
Thanks for your suggestion, and we added these analyses into the discussion. As we all know that the analysis of genotypic classifications is particularly useful for exploring hypotheses that might be formulated about possible causative or preventive factors, as well as for planning health services and making public health decisions.
Point 4: Temperature resistance of the strains has been shown to exist, however, the relevance of the phenotype to the recombination or substitutions have not been discussed therefore it seems irrelevant to the whole manuscript. Although it has been discussed in the conclusion section the relevance of the temperature resistance to the pathogenicity of the virus, but it cannot be attributed to the specific genotype since the changes might be minor to render this resistance.
Response 4: Thanks for your advice, and we added the mutations in RGD in results of 3.6. and the analysis of recombination in discussion in our article. The temperature sensitivity test is a good indicator of virulence and contagiousness. (Sanders BP, de Los Rios Oakes I, van Hoek V, Bockstal V, Kamphuis T, Uil TG, Song Y, Cooper G, Crawt LE, Martín J, Zahn R, Lewis J, Wimmer E, Custers JH, Schuitemaker H, Cello J, Edo-Matas D. Cold-Adapted Viral Attenuation (CAVA): Highly Temperature Sensitive Polioviruses as Novel Vaccine Strains for a Next Generation Inactivated Poliovirus Vaccine. PLoS Pathog. 2016 Mar 31;12(3): e1005483. doi: 10.1371/journal.ppat.1005483.). Because previous studies have shown that the arginine-glycine-aspartic (RGD) motif found in the VP1 capsid protein of CVA9 has a role in cell entry. Analysis of the 19 CVA9 isolates found that all of them possessed the RGD motif (Figure 1). Several different mutations around the RGD motif were seen in strains with different genotypes and temperature sensitivity.
Figure1. CVA9 RGD binding site analysis showing fully conserved nature of the RGD motif in the four representative strains. VP1 position 290 in the protype strain Griggs is equivalent to position 11 in the alignment.
And I agree with your statement you said that although it has been discussed in the conclusion section the relevance of the temperature resistance to the pathogenicity of the virus, but it cannot be attributed to the specific genotype since the changes might be minor to render this resistance. We understand that though the stain China_XJ_2018_99 was classified as F genotype, but there is just one strain was selected to do this analysis, there may be bias to say it attributed to the specific genotype. If we want to account for differences in temperature sensitivity across genotypes, more experimental data from representative strains in different genotypes may be needed.
Point 5:In general, the manuscript gives a good impression on the recombinant events in CAV9 and the migration of the infections between Australia and the USA and Canada. However, the issue of the evolutionary clock for the virus is based on the P1 region which might be the function of an immune pressure which can be variable during time and geographical regions based on epidemics history. Also, the issue of temperature resistance could be due to mutation in the genome or recombination which has not been discussed int the manuscript. Additionally, the whole study is based on the availability of the isolates related to the symptomatic diseases and the active surveillance to detect the virus in population simply might mean active case detection which could bias the whole study and might be good to be acknowledged.
Response 5:Thanks for your advice, we found that the differences in evolutionary rate and time of origin were not particularly large by comparing the phylogeographic analysis of partial VP1 sequences(723nt) and P1 region sequences(2607nt)(Table 1). In order to analyze the dynamic transmission pathways among more countries, we finally selected partial VP1 region sequences for phylogeographic analysis, adding sequences from Japan, Tunisia and Romania, and the maximum clade credibility (MCC) tree also confirms the correctness of the genotypic classification method in phylogenetic analysis.
Substitution Rate (10-3subs./site/year) * |
|
tMRCA(year)# |
||
VP1 |
P1 |
|
VP1 |
P1 |
3.27 (2.93-3.60) |
3.53 (2.54-4.31) |
|
1922 (1911-1932) |
1919 (1881-1941) |
Note: *Estimated mean substitution rates (95%HPD range in parentheses).
#Estimated year of tMRCA (95% CI in parentheses).
And we added the discussion about the issue of temperature resistance could be due to mutation in the genome or recombination in the manuscript. The temperature-insensitive strain had amino acid substitutions at sites T283S, V284M and R288K in the VP1 region, and the temperature-sensitive strain had different acids translated at these three sites. We speculate that the amino acid substitutions at these three sites may be related to the temperature tolerance of the virus, but further animal experiments are needed to verify this.
Thanks for your suggestion, and we admitted that the whole study is based on the availability of the isolates related to the symptomatic diseases and the active surveillance to detect the virus in population simply might mean active case detection which could bias the whole study, and we added the statement in our article. Due to the stable polio-free status of the world, the reduction in laboratory capacity has a knock-on effect of capability to detect and characterize non-polio enteroviruses in samples obtained from patients with neurological symptoms. The development is of concern given the widespread circulation of non-polio enteroviruses, their role as the most common cause of meningitis worldwide, and their involvement in other severe neurological conditions, such as acute flaccid myelitis and encephalitis. So, we should strengthen the monitoring of these non-poliovirus enteroviruses and use a variety of emerging technologies to improve the detection rate in samples in order to identify those that may cause outbreaks as early as possible.
